# The seventh pandemic of cholera in Europe revisited by microbial genomics

Mihaela Oprea[1,9], Elisabeth Njamkepo[2,9], Daniela Cristea[1,9], Anna Zhukova[3,4,9], Clifford G. Clark[5], Anatoly N. Kravetz[6], Elena Monakhova[7], Adriana S. Ciontea[1], Radu Cojocaru[8], Jean Rauzier[2], Maria Damian[1], Olivier Gascuel[3], Marie-Laure Quilici[2] & François-Xavier Weill [2✉]

In 1970, the seventh pandemic of cholera (7 P) reached both Africa and Europe. Between 1970 and 2011, several European countries reported cholera outbreaks of a few to more than 2,000 cases. We report here a whole-genome analysis of 1,324 7 P *V. cholerae* El Tor (7 PET) isolates, including 172 from autochthonous sporadic or outbreak cholera cases occurring between 1970 and 2011 in Europe, providing insight into the spatial and temporal spread of this pathogen across Europe. In this work, we show that the 7 PET lineage was introduced at least eight times into two main regions: Eastern and Southern Europe. Greater recurrence of the disease was observed in Eastern Europe, where it persisted until 2011. It was introduced into this region from Southern Asia, often circulating regionally in the countries bordering the Black Sea, and in the Middle East before reaching Eastern Africa on several occasions. In Southern Europe, the disease was mostly seen in individual countries during the 1970s and was imported from North and West Africa, except in 1994, when cholera was imported into Albania and Italy from the Black Sea region. These results shed light on the geographic course of cholera during the seventh pandemic and highlight the role of humans in its global dissemination.

[1] Cantacuzino National Medico-Military Institute for Research and Development, 050096 Bucharest, Romania. [2] Institut Pasteur, Unité des Bactéries Pathogènes Entériques, Paris 75015, France. [3] Unité Bioinformatique Evolutive, USR3756 (C3BI/DBC), Institut Pasteur & CNRS, Paris, France. [4] Hub Bioinformatique et Biostatistique, USR3756 (C3BI/DBC), Institut Pasteur, Paris, France. [5] National Microbiology Laboratory, Public Health Agency of Canada, Winnipeg, MN, Canada. [6] Kiev Research Institute of Epidemiology and Infectious Diseases, Protasiv Yar Uzuiz, Ukraine. [7] Rostov-on-Don Research Anti-Plague Institute, Rostov-on-Don, Russia 344002. [8] National Centre for Public Health, Chisinau, Republic of Moldova. [9] These authors contributed equally: Mihaela Oprea, Elisabeth Njamkepo, Daniela Cristea, Anna Zhukova. ✉email: francois-xavier.weill@pasteur.fr

Cholera is a life-threatening epidemic diarrheal disease caused by *Vibrio cholerae*. This bacterium is a natural inhabitant of coastal, estuarine and brackish water environments globally[1]. However, a limited number of *V. cholerae* populations are pathogenic for humans, only one of which is associated with the current cholera pandemic[1]. This choleragenic population possesses particular features: it belongs to serotype O1 or, more rarely, serotype O139, and it contains a repertoire of pathogenicity islands and virulence genes, including the CTXɸ prophage, which encodes the cholera toxin (CT) responsible for most of the symptoms of the cholera diarrheal syndrome (i.e., a rapid and massive loss of body fluids leading to severe dehydration), and VPI-1, which contains the genes encoding the toxin-coregulated pilus (TCP) required for adhesion to the intestinal wall and the regulatory gene *toxT* promoting expression of both CT and TCP[2]. In 1970, nine years after its onset in Indonesia, the seventh pandemic of cholera (7P) reached both Africa and Europe[3]. However, little is known about the routes of cholera transmission in Europe during the 7P, beyond the spatiotemporal distribution of cases notified by individual countries to the World Health Organization (WHO)[4]. Furthermore, the available data must be analyzed with care, due to possible underreporting, or even a total absence of reporting, in some countries. For example, in Romania, due to the political situation prevailing before the revolution, no cases of cholera were officially reported to the WHO until 1989, whereas 712 cases in total were reported from 1990 to 1995 (ref. [4]). In the pre-genomic era, only a few attempts were made to understand the population structure, relatedness, and origins of the *V. cholerae* O1 strains implicated in outbreaks in Europe, and even then, only at the level of an individual country or region[5–7]. The typing methods used had a low resolution and were unsuitable for phylogenetic inference. Unsurprisingly, they therefore proved inadequate or even yielded confusing results, particularly for studies of isolates collected over long periods of time. Hence, 51% of the 546 *V. cholerae* O1 isolates collected in Romania between 1977 and 1995 were of the same phage-type, M4, which was identified throughout the study period[5]. By contrast, a DNA-based method, ribotyping (ribosomal RNA gene restriction pattern analysis), performed on 94 toxigenic *V. cholerae* O1 isolates from Romania and Moldova collected between 1977 and 1994, showed these isolates to belong to up to 18 different ribotypes[6].

Large-scale whole-genome sequencing studies have greatly improved our understanding of the population structure, evolution, and spread of *Vibrio cholerae* O1 (refs. [1,8–11]). It has been clearly established that a single lineage, "7PET", is responsible for the current pandemic[1,8]. This lineage has become established in the Bay of Bengal, whence it can spread to the rest of the world[8]. Long-range spread to China[9], Africa[10] or America[1] is well documented by genomic studies. However, these recent studies included only a very small number of European isolates. As a means of elucidating the spatiotemporal spread of cholera in Europe and shedding light on the dynamics of its acquisition of antibiotic resistance through microbial genomics, we selected and sequenced additional *V. cholerae* O1 isolates representative of the widest possible temporal and geographic distribution of European autochthonous cholera cases reported to the WHO[4] or in published studies (Table 1). By contrast with previous findings, we show unambiguously that the European cholera outbreaks studied were caused by repeatedly introduced 7PET populations originating from South Asia and not by local long-term persistent 7PET or non-7PET populations. We also show that the local environment was not involved in the accumulation of antibiotic resistance determinants in cholera outbreak strains over time.

## Results and discussion

We selected 156 *V. cholerae* O1 isolates from European autochthonous cholera cases (Supplementary Data 1). We subsequently found that 12 of these isolates were actually from imported cases, or even from cases occurring in Africa and Asia. They were nevertheless included, to enrich the global phylogenetic context. We contextualized these sequenced *V. cholerae* O1 isolates within a global collection of 1168 7PET genomic sequences (Supplementary Data 1) and constructed a maximum likelihood phylogeny of 1324 genomes—including 172 European genomes—using 13,544 single-nucleotide variants (SNVs) evenly distributed over the non-repetitive, non-recombinant core genome (Supplementary Fig. 1). A time-scaled phylogeny was also constructed from 1283 of these 7PET genomic sequences (Fig. 1).

**Phylogenomics of the 7PET European isolates**. Our phylogenetic analyses clearly showed that eight 7PET sublineages were imported into Europe between 1970 and 2011, leading to the establishment of local transmission chains (Fig. 1). These 7PET sublineages, encompassing the three genomic waves described by Mutreja et al.[8], were named in accordance with a standardized nomenclature (Table 2). In 1970, an "Ogawa strain"—named at the time according to its serotype—was first reported between July and October 1970, in cholera cases in the Black Sea region (Odessa and Kerch) of the former USSR, the former Czechoslovakia, the Middle East (Turkey, Lebanon, and Israel), North Africa (Libya and Tunisia), and West Africa (Guinea, Sierra Leone, Liberia, Ghana, and Côte d'Ivoire)[3]. This "Ogawa strain" corresponds to our sublineage EUR1/AFR1. This sublineage continued to circulate in Africa until the mid-1990s[10]. All but one of the 22 European isolates collected in the early 1970s and studied here belonged to sublineage EUR1/AFR1. These isolates include four Eastern European isolates collected in 1970 that clustered together. In particular, all these isolates had a specific synonymous SNV (T → C) at nucleotide position 2955669 according to GenBank accession number LT907989. One was collected from Odessa in the Black Sea region (now Ukraine) and three were collected from the eastern part of former Czechoslovakia (now Slovakia). The first of these isolates is a representative of the cholera outbreak that began in Odessa in August 1970, with more than 300 cases reported for this year in Ukraine[12]. The other three isolates are representative of a small outbreak (four cases and one death were reported to the WHO but underreporting by the authorities is suspected) that occurred in late October around Vojany, among workers involved in the building of a power plant[13]. Our genomic analysis suggests that the disease was probably exported from Ukraine to Vojany, which is located only 15 km from the Western Ukrainian border, rather than being caused by the effluent of aircraft flying from regions of endemic cholera in India to Europe, as proposed by Rondle et al.[14]. One hypothesis put forward to explain the origin of an unexpected cholera outbreak in Guinea, West Africa, in the summer of 1970 was the return of Guinean students from the Black Sea coast[15]. The presence of a specific synonymous SNV (T → C) at nucleotide position 2955669 in our Ukrainian and Slovakian isolates (but not in the African isolates) is not consistent with this hypothesis and instead suggests the involvement of students, pilgrims or soldiers flying back from the Middle East[3]. The five isolates collected in 1971 in Portugal and Spain, which reported 64 and 22 non-imported cases, respectively, for this year, did not group together by country of isolation in the phylogenetic analysis. They belonged to the upper part of the EUR1/AFR1 sublineage, in which most of the isolates collected after 1970 were from North Africa. This suggests that the

**Table 1 Main cholera outbreaks reported by European countries, except for the Russian Federation, according to the World Health Organization (WHO).**

| Country | Year | Cases (n) | Deaths (n) | Sequenced isolates (n) | 7PET wave[a] | 7PET sublineage[b] |
|---|---|---|---|---|---|---|
| Southern Europe | | | | | | |
| Portugal | 1971 | 64 | 4 | 3 | 1 | EUR1/AFR1 |
| | 1974 | 2467 | 48 | 5 | 1 | EUR1/AFR1 |
| | 1975 | 1066 | 8 | 2 | 1 | EUR1/AFR1 |
| Spain | 1971 | 22 | ? | 2 | 1 | EUR1/AFR1 |
| | 1979 | 267 | ? | 0 | ? | ? |
| Italy | 1973 | 278 | 23 | 2 | 1 | EUR1/AFR1 |
| | 1994 | 12 | 9[c] | 1 | 2 | EUR6/AFR8 |
| Albania | 1994 | 626 | 25 | 3 | 2 | EUR6/AFR8 |
| Eastern Europe | | | | | | |
| Czechoslovakia | 1970 | 4 | 1 | 3 | 1 | EUR1/AFR1 |
| Romania[d] | 1990 | 270 | 1 | 40 | 2 | EUR4/AFR6 |
| | 1991 | 226 | 9 | 25 | 2 | EUR5 |
| | 1994 | 80 | 4 | 22 | 2 | EUR6/AFR8 |
| | 1995 | 118 | 3 | 23 | 2 | EUR7 |
| Republic of Moldova[e] | 1995 | 240 | 5 | 5 | 2 | EUR6/AFR8 |
| Ukraine[e] | 1991 | 75 | 0 | 0 | ? | ? |
| | 1994 | 813 | 20 | 10 | 2 | EUR6/AFR8 |
| | 1995 | 525 | 10 | 2 | 2 | EUR6/AFR8 |
| | 2011 | 33 | 10 | 3[f] | 3 | EUR8 |

[a]Unknown; According to Mutreja et al.[8].
[b]Seventh pandemic Vibrio cholerae El Tor (7PET) sublineages according to our study.
[c]No death according to Maggi et al.[23].
[d]Data reported to the WHO from 1990.
[e]Data reported to the WHO from 1991.
[f]Sequenced by Kuleshov et al.[24] and Smirnova et al.[25].

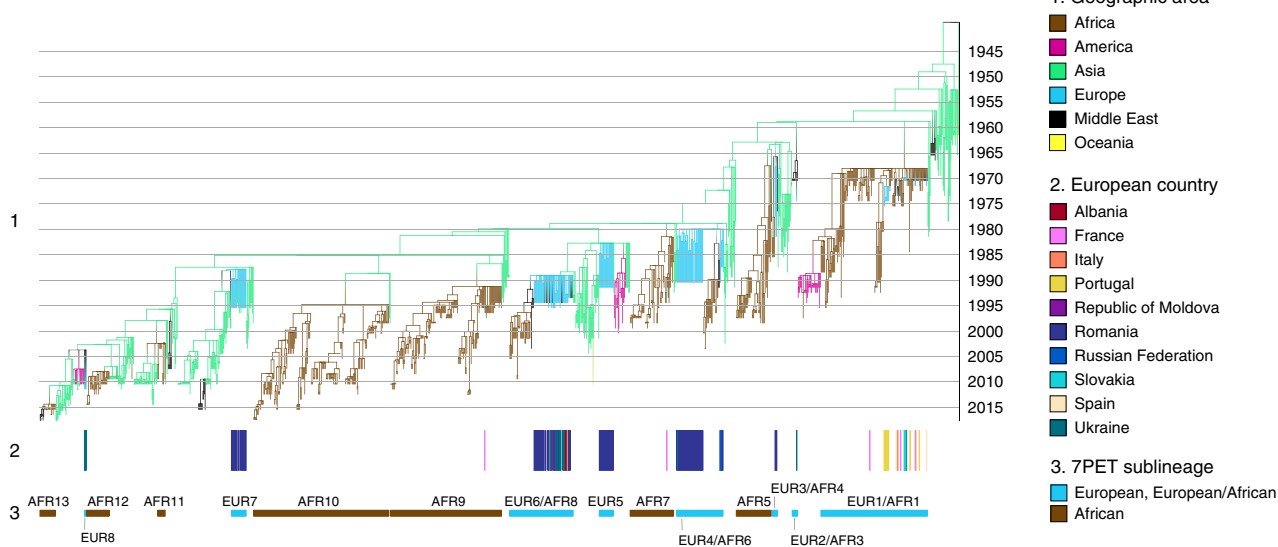

**Fig. 1 Timed phylogeny of the seventh pandemic _V. cholerae_ O1 El Tor isolates analyzed here. A6 was used as an outgroup.** Using the 1,324 seventh pandemic _V. cholerae_ O1 El Tor (7PET) genomic sequences, we identified 41 isolates as outliers (Supplementary Data 1), which were removed before the construction of this dated tree. (1) Branches are colored according to the reconstruction of the ancestral geographic locations of the isolates. Below the phylogeny (2) indicates the countries of origin of the European isolates and (3) indicates the 7PET sublineages concerned.

relatively small number of scattered cholera cases reported by these two countries in 1971 may have been due to multiple importations of the bacterium at a time at which the disease was active in the southern part of the Mediterranean basin. Algeria, for example, reported 1332 cholera cases in 1971 (refs. [4,16]). Unfortunately, no Algerian isolates from 1971 were available for this study. Despite being speculative, our hypothesis may be supported by the occurrence, in the summer of 1971, of a cluster

of seven cholera cases in two villages of Zaragoza province, located on a route used by immigrant workers from North Africa to cross Spain[17]. In 1973, Italy reported 278 cholera cases and 23 deaths from this disease between August and October. Explosive outbreaks occurred around major ports in three Southern provinces: Campania, Apulia, and Sardinia[18]. The two isolates studied here, one from Campania and one from Apulia, clustered together and with Tunisian isolates collected in 1973 and 1974 in

**Table 2 Standardized nomenclature of the seventh pandemic *Vibrio cholerae* El Tor sublineages introduced into Europe, Africa, and Americas.**

| Previous nomenclatures from: | | | Standardized nomenclature |
|---|---|---|---|
| Weill et al.[10] | Domman et al.[1] | Weill et al.[11] | |
| T1 | | | EUR1/AFR1 |
| T2 | LAT-1 | | AFR1/LAM1 |
| T3 | | | EUR2/AFR3 |
| T4 | | | EUR3/AFR4 |
| T5 | | | AFR5 |
| T6 | | | EUR4/AFR6 |
| T7 | | | AFR7 |
| T8 | | | EUR6/AFR8 |
| T9 | | | AFR9 |
| T10 | | | AFR10 |
| T11 | | | AFR11 |
| T12 | | | AFR12 |
| | LAT-2 | | LAM2 |
| | LAT-3 | | LAM3 |
| | | T13 | AFR13 |
| | | | EUR5[a] |
| | | | EUR7[a] |
| | | | EUR8[a] |

*AFR* Africa, *EUR* Europe, *LAM* Latin America.
[a]Sublineages described in this study.

the phylogenetic analysis. All these isolates had a specific synonymous SNV (T → C) at nucleotide position 1462272. In 1973, Tunisia, which is 85 miles away from Sicily, reported 656 cholera cases, up from four in the preceding year[4]. The largest European outbreak during the seventh pandemic was that in Portugal. This country reported 2467 cases and 48 deaths from cholera between April and October 1974 (refs. [4,19]). The disease spread from the south to the north of the country over a period of 20 days, with 17 of the 18 administrative areas in Portugal eventually reporting cases of cholera caused by *V. cholerae* O1 serotype Inaba. We found that the Portuguese isolates from 1974 belonged to sublineage EUR1/AFR1, but were not derived from the strains circulating in this country in 1971. Instead, the 1974 isolates were derived from Angolan strains isolated in 1971 and 1972. Interestingly, all but two of the 10 Portuguese and Angolan isolates sequenced had a non-synonymous SNV (G103A; alteration C03 according to the nomenclature of ref. [10]) in the *wbeT* gene associated with the Inaba serotype (Supplementary Data 1). The other two isolates, obtained from two members of the same family (one case and one carrier) contaminated in Portugal in 1974, belonged to serotype Ogawa (wild-type *wbeT* gene). Both could be classified as true Ogawa revertants of unknown significance. Our phylogeny is consistent with the stationing of colonial troops in the city of Tavira, where the outbreak began, in 1974 (ref. [19]). These troops had been traveling back and forth between Portugal and its African colonies, such as Angola, where 934 cholera cases were reported in 1974 (more than three times the number of cases observed in 1972 and 1973). In July 1975, the cholera epidemic resumed in Portugal, with 1,066 cases and eight deaths[4]. We were able to identify the strain concerned as the same 7PET strain responsible for the outbreak in the preceding year, but were unable to determine whether this strain had persisted from the 1974 outbreak or had been reimported by the thousands of refugees from Angola that entered Portugal in Summer 1975. The robust phylogenetic framework developed here also enabled us to trace the origins of three sporadic autochthonous cases of cholera reported in France in the

early 1970s. The three isolates concerned—which are detailed in the Supplementary Notes section "French autochthonous sporadic cases"—belonged to EUR1/AFR1 and probably originated from North Africa ($n = 2$) or West Africa ($n = 1$).

In 1970, a second cholera strain was identified, contemporary to the introduction into Europe of the "Ogawa strain" (EUR1/AFR1 sublineage). This "Inaba strain"—also named at the time according to its serotype—was first described in Astrakhan (former USSR) in August, and then in the Middle East and Eastern Africa, before the end of 1970 (ref. [3]). This "Inaba strain" corresponds to our EUR2/AFR3 sublineage, the isolates of which belong to the Inaba serotype, with a premature stop codon in the *wbeT* gene (mutation C157T; alteration A05). This sublineage represents the second introduction of cholera into Europe. The small outbreak (less than 100 cases) of cholera caused by *V. cholerae* O1 serotype Inaba reported by Ukraine[12] in 1974 was found to be caused by EUR2/AFR3 (Supplementary Data 1).

We subsequently identified six other introductions of cholera from South Asia into Europe, all in Eastern Europe, including five involving Romania (Figs. 1 and 2, Tables 1 and 3, and Supplementary Data 1). The last introduction occurred in 2011. In three instances, the introductions of 7PET sublineages into Europe (EUR3/AFR4, EUR4/AFR6, and EUR6/AFR8) involved Romania, but subsequently displayed a broader geographic expansion, with similar patterns of spread. EUR3/AFR4 was the sublineage involved in the third introduction of cholera into Europe, and the first into Romania, where it was found in 1977 and 1981. As no cholera cases were officially reported by Romania during this period (Table 1), we estimated the burden of disease due to this sublineage using available data from Tulcea[20] (Table 4), a county encompassing the Danube delta, which was one of the most severely affected regions from 1990 to 1995, when data were reported to the WHO. These local data indicate that 115 cases were identified in Sulina, a port at the mouth of the Danube on the Black Sea in 1977. In 1981, 154 cholera cases were identified along the Danube River (including its delta), and the disease was thought to have been imported from Constanța, one of the largest cities in Romania and the most important harbor on the Black Sea. EUR3/AFR4 also comprised isolates from Turkey (1976–1980) and was found later in the Horn of Africa in 1985. The EUR4/AFR6 sublineage was responsible for the fourth introduction of cholera into Europe and the second into Romania. This sublineage was found in Romania in both 1987 and 1990. In 1987, the health authorities of Tulcea[20] reported 149 cases of cholera at 37 sites, mostly along the Danube river and its branches. In 1990, Romania provided nationwide data for the first time, with 270 cases of cholera and one death from the disease in July, in the Danube delta area. EUR4/AFR6 was also detected in Russia (Stavropol) in 1990, Turkey in 1991 and in East Africa from 1993 to 2003. One EUR4/AFR6 isolate from Ukraine was obtained from a water sample collected from the Danube in 1990, rather than from a human case. EUR6/AFR8 was responsible for the sixth introduction of cholera into Europe and the fourth into Romania. It was identified in Romania in 1993, when the country reported 15 cholera cases, including one imported case, and again in 1994 (80 reported cases, including 46 imported cases). This sublineage was widespread (Supplementary Fig. 2) and caused relatively large outbreaks in neighboring countries, such as Ukraine in 1994 (813 cases and 20 deaths) and 1995 (525 cases and 10 deaths), and the Republic of Moldova in 1995 (240 cases and 5 deaths). Interestingly, EUR6/AFR8 was detected in a fish isolate collected in Odessa in October 1993, one year before the start of the Ukrainian outbreak (September 1994)[12,21]. In 1994, EUR6/AFR8 spread to Southern Europe: Albania (626 cases and 20 deaths) in September[4,7,22], then Italy in October, with a small cluster of 12 cases (no deaths) in Apulia, close to Albania[4,7,23].

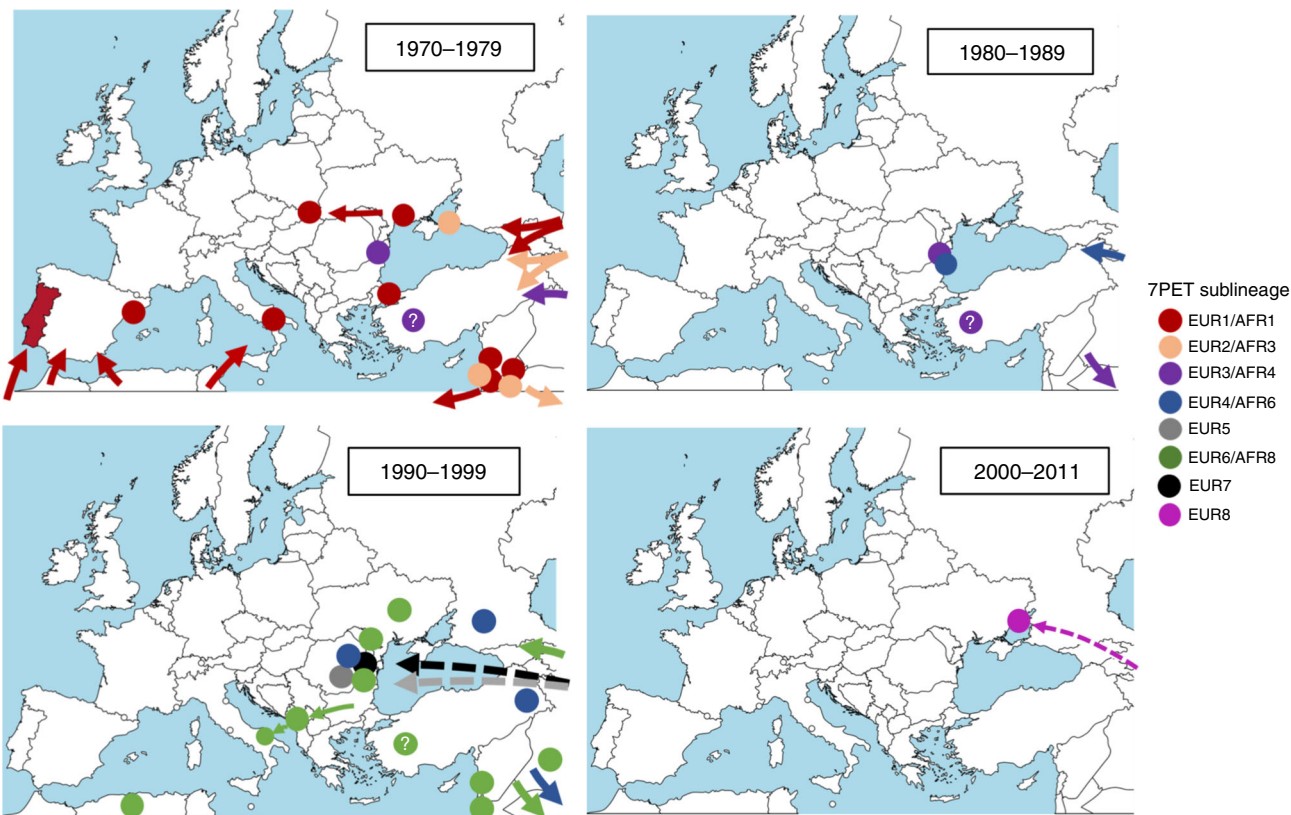

**Fig. 2 Inferred propagation routes of seventh pandemic *V. cholerae* O1 El Tor populations to, from and within Europe between 1970 and 2011.** The circles indicate the location within the country from which outbreak isolates were obtained. A question mark within the circle indicates that the precise site of origin within the country is unknown. The seventh pandemic *V. cholerae* O1 El Tor (7PET) sublineages are indicated by different colors. Arrows, particularly dashed arrows, represent approximate and not precise routes of propagation. The maps were created with mapchart.net (https://mapchart. net/feedback.html).

EUR6/AFR8 was also found in the Middle East (Turkey, Lebanon, Palestine, Iraq) and North Africa (Algeria) between 1993 and 1995, and eventually spread to Eastern and Southern Africa where it circulated until 2009. In three other instances, the introduction was restricted to a single country and year, such as Romania in 1991 (EUR5 sublineage; third introduction of cholera into Romania), Romania in 1995 (EUR7; fifth and last introduction of cholera into Romania) and Ukraine in 2011 (EUR8). The number of cases was moderate, from 58 to 226 (refs. [4,24,25]). Interestingly, EUR7 was not involved in the contemporary outbreaks in Ukraine and Moldova (see above), even though the first cases in 1995 were recorded in two Romanian towns on the border with Ukraine[20].

**Comparison of genomic data with other typing schemes.** Ribotyping was used in previous studies investigating the global population structure of *V. cholerae* O1 (refs. [26,27]) or to investigate 7PET European outbreak isolates[6,7]. In our dataset, ribotyping data were available for 51 isolates from the five sublineages introduced into Romania (Table 5 and Supplementary Data 1). Up to five ribotypes were observed among isolates from the same sublineage (EUR4/AFR6), whereas some ribotypes, such as B5a and B21a, were common to isolates from two or even three sublineages (B5a in EUR3/AFR4 and EUR4/AFR6 and B21a in EUR3/AFR4, EUR5, and EUR6/AFR8). This confirms the irrelevance of non-genomic methods for inferring true genetic relationships between 7PET isolates and for accurate studies of the spatiotemporal spread of 7PET.

**Evolution of antimicrobial resistance in European 7PET isolates.** We show here that even though each sublineage is well characterized by its antimicrobial resistance gene content at genomic level (Table 3), diverse antimicrobial resistance phenotypes are observed within each sublineage (Fig. 3, Table 6 and Supplementary Data 1). Several explanations for this observation can be proposed. Firstly, it may be due to genetic rearrangements involving AMR determinants and affecting a proportion of isolates from a given 7PET sublineage. For example, some isolates from the EUR5 and EUR6/AFR8 sublineages lost their GI-15 genomic island conferring resistance to streptomycin and sulfamethoxazole, and some isolates from the EUR6/AFR8 and EUR7 sublineages presented an internal deletion of their ICE*Vch*Moz10/ICE*Vch*Ban9 and ICE*Vch*Ind5/ICE*Vch*Ban5 genomic islands, respectively (Table 3). This deletion encompassed three to four AMR genes. Secondly, some genes may not confer high levels of resistance in *V. cholerae*. This is the case for *tetA* resistance to tetracycline and for *floR* resistance to phenicols, particularly in tests with chloramphenicol disks. Thirdly, the clinical breakpoints used to define susceptibility, intermediate resistance or resistance in *V. cholerae* may not be entirely appropriate. No breakpoints were determined for this pathogen by the European Committee for Antimicrobial Susceptibility Testing (EUCAST)[28], and those for *Vibrio* spp. given by the Clinical and Laboratory Standards Institute (CLSI)[29,30] are actually derived from data for Enterobacteriaceae. Finally, we cannot rule out the possibility of changes in AST findings for isolates tested retrospectively, after several decades of storage. Given these limitations, antimicrobial susceptibility testing (AST)

**Table 3 Characteristics of the *Vibrio cholerae* O1 El Tor sublineages introduced into Europe, 1970–2011.**

| Sublineage | Wave[a] | Location in Europe | Extension beyond Europe | AMR determinants[b] | | | | | | |
|---|---|---|---|---|---|---|---|---|---|---|
| | | | | *VC_0715* | *VC_A0637* | GI-15 | SXT/R391 element | | *gyrA* | *parC* |
| EUR1/AFR1 | 1 | Ukraine 1970 Slovakia 1970 Spain 1971 Portugal 1971, 1974–1975 Italy 1973 | Middle East Whole of Africa | WT | WT | — | — | | WT | WT |
| EUR2/AFR3 | 1 | Ukraine 1974 | Middle East Eastern Africa | WT | WT | — | — | | WT | WT |
| EUR3/AFR4 | 1 | Romania 1977–1981 | Middle East Eastern Africa | WT | WT | — | — | | WT | WT |
| EUR4/AFR6 | 2 | Romania 1987–1990 | Middle East Eastern Africa | WT | WT | — | — | | WT | WT |
| EUR5 | | Romania 1991 | None | R169C | Q5Stop | GI-15[c] | — | | WT | WT |
| EUR6/AFR8 | 2 | Romania 1993–1994 | Middle East | R169C | Q5Stop | GI-15[c] | ICE*Vch*Ban9[d] | | WT | WT |
| | | Ukraine 1993–1995 Moldova 1995 Albania 1994 Italy 1994 | Northern Africa Eastern Africa Southern Africa | | | | | | | |
| EUR7 | 3 | Romania 1995 | None | R169C | Q5Stop | | ICE*Vch*Ind5[e] | | S83I | WT |
| EUR8 | 3 | Ukraine 2011 | None | R169C | Q5Stop | | ICE*Vch*Ind5 | | S83I | S85L |

AMR antimicrobial resistance, WT wild type, — absence.
[a]According to Mutreja et al.[8].
[b]These non-silent point mutations in chromosomal genes *VC_0715* and *VC_A0637*, and *gyrA* and *parC* are associated with resistance to nitrofurans, and to quinolones, respectively; the GI-15 genomic island harbors the genes responsible for resistance to streptomycin (*aadA*-like) and sulfonamides (*sul1*); the ICE*Vch*Ban9/ICE*Vch*Moz10 genomic island harbors the genes responsible for resistance to streptomycin (*strAB*), tetracycline (*tetA*), chloramphenicol (*floR*), sulfonamides (*sul2*), trimethoprim/vibriostatic agent O/129 (*dfrA1*), and cotrimoxazole (*sul2* and *dfrA1*); the ICE*Vch*Ind5/ICE*Vch*Ban5 genomic island harbors the genes responsible for resistance to streptomycin (*strAB*), chloramphenicol (*floR*), sulfonamides (*sul2*), trimethoprim/vibriostatic agent O/129 (*dfrA1*), and cotrimoxazole (*sul2* and *dfrA1*).
[c]GI-15 is absent in rare isolates.
[d]Frequent ICE*Vch*Ban9 internal deletion encompassing the *strAB*, *tetA*, *floR*, and *sul2* genes.
[e]Rare ICE*Vch*Ind5 internal deletion encompassing the *strAB*, *floR*, and *sul2* genes.

**Table 4 Characteristics of the cholera cases in Romania, 1977–1995.**

| Year | National data[a] cases (incl. imported)/deaths | Data for Tulcea | |
|---|---|---|---|
| | | Cases (incl. carriers)/ deaths | Characteristics |
| 1977 | — | 115 (76)/? | 21 September to 7 October; Sulina |
| 1981 | — | 154 (101)/4 | 17 August to 8 November; towns along the Danube river and delta; imported from Constanța; poor hygiene and drinking of river water |
| 1987 | — | 275 (126)/0 | 21 July to 23 November; Tulcea city then 37 sites along the Danube river and its branches; poor hygiene and drinking of river water |
| 1990 | 270/1 | — | |
| 1991 | 226/9 | 142 (31)/0 | 6 August to 11 October; Badabag then other towns; imported from Turkey; poor hygiene and drinking of unsafe water (river, fountain) |
| 1992 | 3/0 | — | |
| 1993 | 15 (1)/0 | — | |
| 1994 | 80 (46)/4 | — | |
| 1995 | 118/3 | 79 (19)/0 | 4 August to 7 October; Grindu and Ceatalchioi, then downstream along the Danube river; drinking of river water, river activities or interhuman transmission |

— not reported, ? unknown.
[a]Reported to the WHO.

should be used with caution for distinguishing and tracking 7PET strains over time.

Antimicrobial resistance susceptibility testing was previously performed on hundreds of *V. cholerae* O1 isolates collected between 1977 and 1995 in Romania[5,31]. These studies documented a gradual increase in antibiotic resistance, with the acquisition of resistance to nitrofurans in 1990–1991, tetracycline in 1993–1994, cotrimoxazole in 1993–1995 and nalidixic acid in 1995. Our genomic and phenotypic data are largely consistent with both these previous studies. However, in our dataset, the first isolates resistant to nitrofurans—caused by point mutations in the *VC_0715* and *VCA0637* nitroreductase genes[11]—were isolated in 1991 (EUR5 sublineage) and not in 1990 (EUR4/AFR6). Israil et al.[31] suggested that isolates resistant to tetracycline (TET), cotrimoxazole

(SXT), and nalidixic acid (NAL) emerged in Romania following the widespread use of TET and SXT in 1990–1992 and of NAL in 1993–1994 in Romania. Our phylogenomic analysis on contextual global 7PET isolates clearly shows that the main changes in AMR were driven by the introduction of sublineages into Europe, which already carried AMR determinants in South Asia before their introduction into Europe (Supplementary Data 1). This finding makes it possible to reject the hypothesis[31] that the local use of antibiotics in Romania was responsible for the emergence of AMR *V. cholerae* O1 strains.

**Table 5 Correlation between genomic data and ribotypes.**

| Sublineage | Genomic wave[a] | Ribotype[b] (number of isolates) |
|---|---|---|
| EUR3/AFR4 | 1 | B5a (6), B21a (1) |
| EUR4/AFR6 | 2 | B5a (5), B5b (2), B7 (1), B8a (4), B10 (1) |
| EUR5 | 2 | B21a (5) |
| EUR6/AFR8 | 2 | B21a (16), B21b (1), B26 (2) |
| EUR7 | 3 | B27 (7) |

[a]According to Mutreja et al.[8].
[b]According to Damian et al.[6] and Koblavi et al.[26].

**Genomic epidemiology and sources of cholera.** Even though it did not cover all notified outbreaks (Supplementary Notes section "7PET outbreak isolates not studied"), our genomic analysis identified two different patterns in Europe during the seventh pandemic of cholera. The first pattern concerned the Southern European countries, which were hit during the 1970s, when the disease was very active in Africa, particularly in North Africa, which is geographically close to Southern Europe, and West Africa, due to its colonial ties with European countries. The disease was easily brought back to Europe by colonial troops, migrant workers, refugees or tourists. Studies on imported cholera cases in Europe between the mid-1970s and late 1980s showed that most of these cases occurred in citizens of nations in which cholera was endemic (migrant workers or refugees), mostly in North Africa[32,33]. The outbreaks in Southern Europe were of limited size, with the exception of the 1974–1975 outbreak in Portugal. Good hygiene, access to safe water, and good sanitation conditions were probably responsible for the containment of the disease. Secondary cases of cholera are, thus, extremely rare in high-income countries[32,34]. In several of these outbreaks, epidemiological investigations identified raw fish and seafood as the vehicle of transmission[18,19,23]. The larger extent of the cholera outbreak in Portugal in 1974 may be partly due to the contamination of one brand of commercially bottled water[19,35].

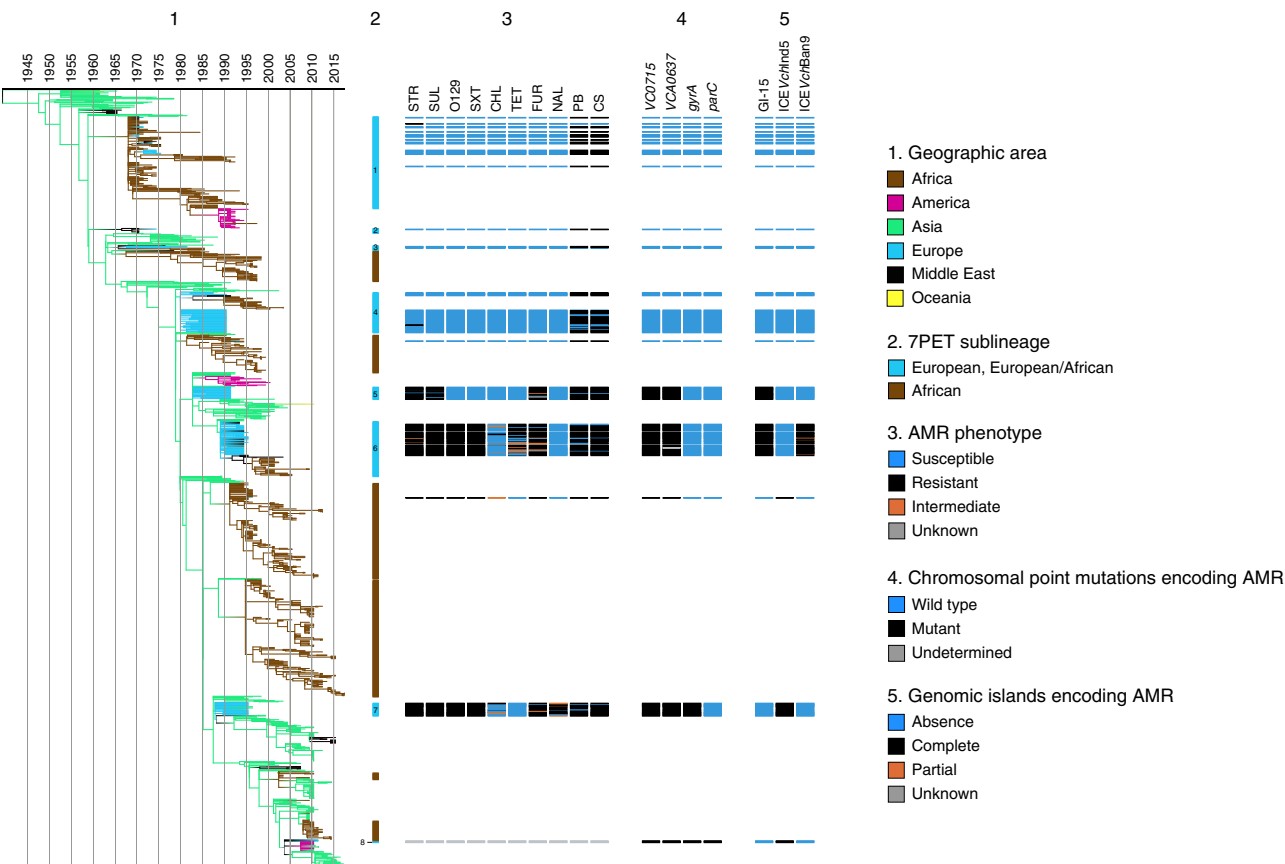

**Fig. 3 Phenotypic and genetic characterization of antibiotic resistance in the European seventh pandemic *V. cholerae* O1 El Tor isolates analyzed here.** (1) Timed phylogeny, identical to that in Fig. 1, but represented vertically. (2) The seventh pandemic *V. cholerae* O1 El Tor (7PET) sublineages are indicated. The names of the European and European/African sublineages are indicated by a one-digit number within or next to the blue line and corresponding to that found in the sublineage name (e.g., 1 for EUR1, 2 for EUR2/AFR3, etc). The antimicrobial resistance (AMR) phenotype for 10 antibiotics, the presence of chromosomal AMR genes (4), and genomic islands encoding AMR (5) are shown for each isolate, according to its position in the phylogeny. More detail about the genes and genomic islands encoding AMR is provided in the footnotes of Table 3. The abbreviations used for the antibiotics are as follows: STR streptomycin; SUL sulfonamides; O129 vibriostatic agent O/129; SXT trimethoprim-sulfamethoxazole; CHL chloramphenicol; TET tetracycline; FUR furazolidone; NAL nalidixic acid; PB polymyxin B; CS colistin. Note that 18 European isolates identified as outliers (Supplementary Data 1) are not represented in this timed phylogeny.

**Table 6 Antimicrobial resistance phenotypes of 167 European *V. cholerae* O1 El Tor isolates, according to their phylogenetic grouping.**

| Sublineage | Genomic wave[a] | AMR phenotypes[b] (number of isolates) |
|---|---|---|
| EUR1/AFR1 | 1 | PB, CS (20) |
| | | STR (1) |
| EUR2/AFR3 | 1 | PB, CS (1) |
| EUR3/AFR4 | 1 | PB, CS (3) |
| | | Pan-susceptible (1) |
| EUR4/AFR6 | 2 | PB, CS (35) |
| | | Pan-susceptible (9) |
| | | CS (1) |
| | | STR (1) |
| EUR5 | 2 | STR, SUL, FUR, PB, CS (16) |
| | | STR, SUL, PB, CS (2) |
| | | FUR, PB, CS (2) |
| | | STR, SUL (2) |
| | | STR, SUL, FUR (1) |
| | | STR, FUR, PB, CS (1) |
| | | STR, SUL, FUR, CS (1) |
| EUR6/AFR8 | 2 | STR, SUL, O129, SXT, TET, FUR, PB, CS (22) |
| | | STR, SUL, O129, SXT, FUR, PB, CS (9) |
| | | STR, SUL, O129, SXT, CHL, TET, FUR, PB, CS (8) |
| | | STR, SUL, O129, SXT, FUR (2) |
| | | STR, SUL, O129, SXT, PB, CS (2) |
| | | STR, SUL, O129, SXT, CS (1) |
| | | STR, SUL, O129, SXT (1) |
| | | STR, SUL, O129, SXT, TET, FUR (1) |
| | | STR, SUL, O129, SXT, TET, FUR, CS (1) |
| EUR7 | 3 | STR, SUL, O129, SXT, FUR, NAL, PB, CS (10) |
| | | STR, SUL, O129, SXT, CHL, FUR, NAL, PB, CS (4) |
| | | STR, SUL, O129, SXT, FUR, PB, CS (3) |
| | | STR, SUL, O129, SXT, CHL, FUR, PB, CS (2) |
| | | O129, FUR, NAL (1) |
| | | STR, SUL, O129, SXT (1) |
| | | STR, SUL, O129, SXT, FUR, NAL, CS (1) |
| | | STR, SUL, O129, SXT, NAL (1) |

*CHL* chloramphenicol, *CS* colistin, *FUR* furazolidone, *NAL* nalidixic acid, *PB* polymyxin B, *STR* streptomycin, *SUL* sulfonamides, *SXT* trimethoprim-sulfamethoxazole, *O129* vibriostatic agent O/129, *TET* tetracycline.
[a]According to Mutreja et al.[8].
[b]AMR antimicrobial resistance, Intermediate resistance (I), and resistance (R) have been grouped together to define the AMR phenotypes.

The second pattern was seen in Eastern European countries bordering the Black Sea, which were recurrently affected, over a longer period of time, by cholera strains originating from South Asia. For example, Romania experienced five introductions of cholera, three of which occurred during a 4-year period in the 1990s (Supplementary Fig. 3). However, most of these introductions were linked to a more global spread of these sublineages involving the Middle East and eventually reaching Africa, including East Africa in particular. Another particular feature of cholera in Eastern Europe was that most of the cases occurred during warmer seasons (July to the end of October) in the Danube delta and along the Black Sea Coast, where the main risk factors for cholera were the drinking of surface waters directly from the Danube (particularly in remote locations or during disruptions at water treatment plants), a lack of food hygiene and the consumption of uncooked or undercooked fish[5,12,19].

Understanding the balance between the role of the aquatic environment as a long-term reservoir of toxinogenic *V. cholerae* and the role played by humans in the spread and maintenance of the pathogen—through direct (human-to-human) or indirect (due to transient pollution of the environment with feces from cholera patients) transmission—is of particular importance if we are to combat this scourge effectively. The "cholera paradigm" theory, which has prevailed for the last two decades, suggests that coastal environmental factors influenced by climate and weather

conditions are the key elements underlying cholera epidemiology[36]. This theory, based on the clear association between *V. cholerae* populations and plankton communities in coastal ecosystems, was developed on the basis of observations made primarily in the Bay of Bengal[36,37], the global hub and natural habitat of the 7P cholera agent. However, outside Asia, genomic reconstruction of the spread of 7PET over several decades was not consistent with perennial aquatic environmental reservoirs acting as the primary source of epidemic cholera in Africa[10] and the Americas[1]. The data presented here—showing that 7PET sublineages, recurrently introduced from Asia, ultimately became extinct after a few years of circulation in Europe—also clearly rule out the long-term establishment of an aquatic reservoir of the disease in Europe, particularly in the countries bordering the Black Sea. Instead, our data suggest that the dynamics of cholera in this region, which houses a number of major ports, may be accounted for by the recurrent importation of 7PET strains via the movement of populations, such as the arrival of migrant workers from areas of endemic cholera during the 1990s[5]. Direct (human-to-human) or indirect (pollution of the environment with feces from cholera patients) secondary transmission can then cause cholera outbreaks in situations of poor hygiene, inadequate sanitation, the unsanitary harvesting and marketing of seafood, and risky dietary and drinking habits. As the movement of populations from regions in which cholera is endemic has not

ceased, and, indeed, has even increased over time, improvements in water, sanitation and hygiene (WASH) services and in health infrastructures almost certainly prevented the occurrence of cholera outbreaks in Europe after 2011.

## Methods

**Bacterial isolates**. The 156 7PET isolates sequenced in this study are listed in Supplementary Data 1. They were obtained from the collections of the French National Reference Center for Vibrios and Cholera, Institut Pasteur, Paris, France; the Cantacuzino NMMIRD; and the Public Health Agency of Canada.

The isolates were characterized by standard biochemical, culture, and serotyping methods[38].

**Antibiotic susceptibility testing**. Antibiotic susceptibility was determined by the disk diffusion method, on Mueller-Hinton (MH) agar, in accordance with the guidelines of the Antibiogram Committee of the French Society for Microbiology (CA-SFM 2006)[39]. The following antimicrobial drugs (Bio-Rad, Marnes-la-Coquette, France) were tested: ampicillin (AMP), cefalotin (CEF), cefotaxime (CTX), streptomycin (STR), chloramphenicol (CHL), azithromycin (AZM), sulfonamides (SUL), trimethoprim-sulfamethoxazole (SXT), vibriostatic agent O/129 (O129), tetracycline (TET), nalidixic acid (NAL), ciprofloxacin (CIP), nitrofurantoin (FUR), polymyxin B (PB) and colistin (CS). *Escherichia coli* CIP 76.24 (ATCC 25922) was used as a control.

**Total DNA extraction**. Total DNA was extracted with the Wizard Genomic DNA Kit (Promega, Madison, WI, USA), the Maxwell 16-cell DNA purification kit (Promega, Madison WI) or the DNeasy Blood & Tissue Kit (Qiagen), in accordance with the manufacturer's recommendations.

**Whole-genome sequencing**. Whole-genome sequencing was carried out at the PF1 sequencing platforms of the Institut Pasteur, the genotyping and sequencing core facility of the Institut du Cerveau et de la Moëlle Epinière (Paris, France), or at the Public Health Agency of Canada, on Illumina platforms generating 70–300 bp paired-end reads, yielding a mean of 99-fold coverage (minimum 16-fold, maximum 1260-fold).

Short-read sequence data were submitted to the European Nucleotide Archive (ENA), under study accession number PRJEB38484; genome accession numbers are provided in Supplementary Data 1.

**Additional genomic data**. Raw sequence files or assembled genomes from 1,168 7PET genomes described by Weill et al.[10,11] or unpublished were downloaded from the ENA or GenBank and included in this study (Supplementary Data 1). We generated 100 bp overlapping simulated reads from the assembled genomes with fasta_to_fastq.pl (https://github.com/ekg/fasta-to-fastq/blob/master/fasta_to_fastq.pl).

**Genomic sequence analyses**. The paired-end reads and simulated paired-end reads were mapped onto the reference genome of *Vibrio cholerae* O1 El Tor N16961 (also known as A19)[8] (GenBank accession numbers LT907989 and LT907990) with Snippy v4.1.0/BWA-MEM v0.7.13 (https://github.com/tseemann/snippy). Single-nucleotide variants (SNVs) were called with Snippy v4.1.0/Freebayes v1.1.0 (https://github.com/tseemann/snippy) under the following constraints: mapping quality of 60, a minimum base quality of 13, a minimum read coverage of 4, and a 75% read concordance at a locus for a variant to be reported. An alignment of core genome SNVs was produced in Snippy v4.1.0 for phylogenetic inference.

Short reads were assembled with SPAdes (ref. [40]) version 3.1.0.

The various genetic markers were analyzed with BLAST[41] version 2.2.26 (https://blast.ncbi.nlm.nih.gov/Blast.cgi) against the reference sequences of the O1 (ref. [10]) and O139 (ref. [42]) *rfb* genes, *ctxB* (ref. [43]), *wbeT* (ref. [10]), VSP-II (ref. [44]), and WASA-1 (ref. [45]).

The presence and type of acquired antibiotic resistance genes (ARGs) or ARG-containing structures were determined with ResFinder[46] version 3.1.0. (https://cge.cbs.dtu.dk/services/ResFinder/), BLAST analysis against GI-15 (ref. [47]), Tn7 (ref. [10]), and SXT/R391 integrative and conjugative elements[48], and PlasmidFinder[49] version 2.0.1. (https://cge.cbs.dtu.dk/services/PlasmidFinder/). The presence of mutations in genes encoding resistance to quinolones (*gyrA*, *parC*)[10], resistance to nitrofurans (*VC_0715*, *VC_A0637*)[11], or restoring susceptibility to polymyxin B (*vprA*)[11] was investigated by analysis of the sequences assembled de novo with BLAST.

The GenBank accession numbers used to perform these analyses are shown in Supplementary Table 1. The data were entered into an Excel (Microsoft) version 15.41 spreadsheet (Supplementary Data 1).

**Phylogenetic analysis**. Repetitive (insertion sequences and the TLC-RS1-CTX region) and recombinogenic (VSP-II) regions in the alignment were masked[10]. Putative recombinogenic regions were detected and masked with Gubbins[50] version 2.3.4. A maximum likelihood (ML) phylogenetic tree was built from an alignment of 13,544 chromosomal SNVs, with RAxML[51] version 8.0.20, under the GTR model, with 200 bootstraps. The final tree was rooted on the A6 genome—the

earliest and most ancestral[10] seventh pandemic isolate, collected in Indonesia in 1957—and visualized with iTOL[52] version 5 (https://itol.embl.de) (Supplementary Fig. 1). The rooted tree was time-scaled with LSD2 (ref. [53]) version 1.6.5, using a strict molecular clock model with temporal outlier detection. LSD2 detected 41 temporal outliers (listed in Supplementary Data 1), which were removed from the time-scaled tree. The ancestral characters for geographic regions were reconstructed on the time-scaled tree with PastML[54] version 1.9.29.2, with the MPPA method and model F81. We ensured that the geographic predictions of the most recent common ancestors (MRCAs) of European clades were not disturbed by biased sampling, by performing reconstruction under the same settings for five trees obtained by pruning some of the leaves corresponding to overrepresented regions from the full time-tree. For each pruned subtree, we retained two (randomly selected) of 36, two of 20, and two of 40 leaves from those sampled in Romania in 1990, 1991, and 1993–1995, respectively, four of 18 leaves from those sampled in Ukraine or Moldova in 1993–1995, all the leaves sampled in India in 1990 ($n = 3$) and 1991 ($n = 3$), and all the leaves sampled from India and Pakistan in 1993–1995 ($n = 9$). This procedure made it possible to bring the sampling proportions for these countries closer to the proportions of reported cases. However, these sampling variations did not change the geographic prediction for the MRCAs of EUR4/AFR6, EUR5, EUR6/AFR8. The time-scaled tree with ancestral geographic reconstruction was visualized with iTOL (Fig. 1).

**Reporting summary**. Further information on research design is available in the Nature Research Reporting Summary linked to this article.

## Data availability

Short-read sequence data were submitted to the European Nucleotide Archive (ENA), under study accession number PRJEB38484, and the genome accession numbers are provided in Supplementary Data 1. The whole-genome alignment for the 1,324 genomes and other files that support the findings of this study have been deposited in FigShare [https://doi.org/10.6084/m9.figshare.5422912]. The phylogeny and associated metadata have been uploaded to Microreact for interactive viewing [https://microreact.org/project/choleraeurope]. The GenBank accession numbers used for our analyses are shown in Supplementary Table 1.

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

## Acknowledgements

This study was supported by the Institut Pasteur, Cantacuzino NMMIRD, Santé publique France, the Fondation Le Roch-Les Mousquetaires, the French government's Investissement d'Avenir programme, Laboratoire d'Excellence 'Integrative Biology of Emerging Infectious Diseases' (grant number ANR-10-LABX-62-IBEID), and "Institut Convergence for the study of Emergence of Pathology Through Individuals and Populations" (INCEPTION, grant number PIA/ANR-16-CONV-0005). D.C. was supported by a traineeship grant from the Calmette and Yersin Program of the Institut Pasteur International Network. M.O. was supported by a mobility grant from the Romanian Ministry of Research and Innovation CNCS - UEFISCDI, project no. 495/2018, within PNCDI III. The Institut Pasteur Biomics platform is supported by France Génomique (ANR10-INBS-09-08) and IBISA. We thank I. Najjar and L. Lemée for sequencing the isolates.

## Author contributions

M.-L.Q. and F.-X.W. designed the study. F.-X.W. oversaw the project. D.C., C.G.C., A.K.K., E.M., R.C., M.D., and M.-L.Q. collected, selected, and provided characterized isolates or genomic sequences and their corresponding epidemiological information. E.N., D.C., J.R. performed the phenotypic experiments and DNA extractions. The genomic sequence data were analyzed by M.O., A.Z., E.N., O.G., and F.-X.W. The paper was written by F.-X.W. and all authors contributed to the editing of the paper.

## Competing interests

The authors declare no competing interests.
