## [Peer Review File · Nature Communications]

REVIEWER COMMENTS

Reviewer #1 (Remarks to the Author):

This paper by Oerpa et al., reports the analysis of 1325 isolates of *Vibrio cholerae* associated with cholera outbreaks in Europe between 1970-2011 using whole genome sequences. The paper claims the most complete analysis of *V. cholerae* isolates found in Europe during this period and established their high relatedness to the 7PET 7th pandemic El Tor lineage. The analysis also showed that cholera was introduced into Europe at least 8 times and the most likely paths for its introduction and spread geographically within Europe. Most importantly the authors conclude "there is no evidence for a permanent aquatic reservoir of the disease in this region" and this is a very important confirmation of other genomic studies that long term in a stable aquatic niche once introduced into a new geographical location. Rather human (travelers, population migrations, and workers) are the most likely sources of these outbreak strains and infected humans are more responsible for the cholera outbreaks by human-to-human or through poor sanitation by environment contamination with human feces followed by unsanitary collection and consumption of seafoods or water products.

I think this study was technically well done and worthy of consideration for publication in Nature Comm. It will continue to build a strong case in the field that the 'environmental hypothesis' for *V. cholerae* emergence in new geographical locations is flawed and further influence decision makers who hope to eradicate this disease through vaccination efforts.

Suggestions to authors:

For the general reader, a bit of background about *V. cholerae* and its emergence as a pathogen might be helpful in the introduction. That the species includes environmental strains but all pathogenic strains carry unique features that include a phage that encodes cholera toxin and an island that encodes a colonization factor. Etc.

Again, for the general reader, a sentence or two discussing the environmental hypothesis for its de novo emergence in new geographical locations should be discussed. There was plenty of controversy about this topic in the context of the Haiti epidemic.

Reviewer #2 (Remarks to the Author):

The authors present the latest in a line of groundbreaking *Vibrio cholerae* genomics studies they have performed, this time identifying the routes of introgression of the 7th pandemic into Europe. As ever the data presented and the analysis performed is to a very high standard and provides a fascinating historical account of the 7th pandemic's path through Europe. Below are some comments I would ask the authors to consider

Major comments:

p4: Line 93: I think a slightly more descriptive wording is required rather than just "our results contrast"

p6: Line 143 and 154: I have no issue with these potential routes of infection being discussed, but it needs to be clear they are purely informed conjecture

p8: Line 188: I really think you need some form of wording supporting the statement of African origin here

p10: I am rather unconvinced that we really still need to be confirming that simple genotyping

methods are not discriminatory in the light of genomic data

p11: I cant help but feel the AMR section would be strengthened by presenting a comprehensive genomic analysis of AMR genes in all the European strains and how this aligns with AST. This could be presented in a single annotated phylogeny

p31: On a personal level I would really like to see a classical rectangular visualisation of this timed phylogeny. I also wonder if some of the national outbreaks need amplified phylogenies as multi panel or supplementary figures to allow the reader to interrogate

Minor comments:

p8: Line 190-206: I personally dont like this style of short paragraphs and would like to see this merged to a single narrative paragraph

p10: Similarly the break in paragraphs at line 240 seems unnecessary

Authors: We would like to thank the reviewers for their many insightful comments. We address these comments, point by point, below.

Reviewer #1 (Remarks to the Author):

This paper by Oerpa et al., reports the analysis of 1325 isolates of *Vibrio cholerae* associated with cholera outbreaks in Europe between 1970-2011 using whole genome sequences. The paper claims the most complete analysis of *V. cholerae* isolates found in Europe during this period and established their high relatedness to the 7PET 7th pandemic El Tor lineage. The analysis also showed that cholera was introduced into Europe at least 8 times and the most likely paths for its introduction and spread geographically within Europe. Most importantly the authors conclude "there is no evidence for a permanent aquatic reservoir of the disease in this region" and this is a very important confirmation of other genomic studies that long term in a stable aquatic niche once introduced into a new geographical location. Rather human (travelers, population migrations, and workers) are the most likely sources of these outbreak strains and infected humans are more responsible for the cholera outbreaks by human-to-human or through poor sanitation by environment contamination with human feces followed by unsanitary collection and consumption of seafoods or water products.

I think this study was technically well done and worthy of consideration for publication in Nature Comm. It will continue to build a strong case in the field that the 'environmental hypothesis' for *V. cholerae* emergence in new geographical locations is flawed and further influence decision makers who hope to eradicate this disease through vaccination efforts.

Suggestions to authors:

For the general reader, a bit of background about *V. cholerae* and its emergence as a pathogen might be helpful in the introduction. That the species includes environmental strains but all pathogenic strains carry unique features that include a phage that encodes cholera toxin and an island that encodes a colonization factor. Etc.

Authors: We have now added this background (lines 63-73).

*"This bacterium is a natural inhabitant of coastal, estuarine and brackish water environments globally¹. However, a limited number of *V. cholerae* populations are pathogenic for humans, only one of which is associated with the current cholera pandemic¹. This cholerae population possesses particular features: it belongs to serotype O1 or, more rarely, serotype O139, and it contains a repertoire of pathogenicity islands and virulence genes, including the CTX ϕ prophage, which encodes the cholera toxin (CT) responsible for most of the symptoms of the cholera diarrheal syndrome (i.e., a rapid and massive loss of body fluids leading to severe dehydration), and VPI-1, which contains the genes encoding the toxin-coregulated pilus (TCP) required for adhesion to the intestinal wall and the regulatory gene *toxT* promoting expression of both CT and TCP²."*

Again, for the general reader, a sentence or two discussing the environmental hypothesis for its de novo emergence in new geographical locations should be discussed. There was plenty

of controversy about this topic in the context of the Haiti epidemic.

Authors: We now discuss the environmental theory (lines 327 to 339).

*“Understanding the balance between the role of the aquatic environment as a long-term reservoir of toxinogenic *V. cholerae* and the role played by humans in the spread and maintenance of the pathogen — through direct (human-to-human) or indirect (due to transient pollution of the environment with feces from cholera patients) transmission — is of particular importance if we are to combat this scourge effectively. The “cholera paradigm” theory, which has prevailed for the last two decades, suggests that coastal environmental factors influenced by climate and weather conditions are the key elements underlying cholera epidemiology³⁶. This theory, based on the clear association between *V. cholerae* populations and plankton communities in coastal ecosystems, was developed on the basis of observations made primarily in the Bay of Bengal^{36,37}, the global hub and natural habitat of the 7P cholera agent. However, outside Asia, genomic reconstruction of the spread of 7PET over several decades was not consistent with perennial aquatic environmental reservoirs acting as the primary source of epidemic cholera in Africa¹⁰ and the Americas¹. The data presented here — showing that 7PET sublineages, recurrently introduced from Asia, ultimately became extinct after a few years of circulation in Europe — also clearly rule out the long-term establishment of an aquatic reservoir of the disease in Europe, particularly in the countries bordering the Black Sea.”*

Reviewer #2 (Remarks to the Author):

The authors present the latest in a line of groundbreaking *Vibrio cholerae* genomics studies they have performed, this time identifying the routes of introgression of the 7th pandemic into Europe. As ever the data presented and the analysis performed is to a very high standard and provides a fascinating historical account of the 7th pandemic's path through Europe. Below are some comments I would ask the authors to consider

Major comments:

p4: Line 93: I think a slightly more descriptive wording is required rather than just "our results contrast"

Authors: We have added the following “ By contrast with previous findings, we show unambiguously that the European cholera outbreaks studied were caused by repeatedly introduced 7PET populations originating from South Asia and not by local long-term persistent 7PET or non-7PET populations. We also show that the local environment was not involved in the accumulation of antibiotic resistance determinants in cholera outbreak strains over time.”

p6: Line 143 and 154: I have no issue with these potential routes of infection being discussed, but it needs to be clear they are purely informed conjecture

Authors: We have added the following, on lines 159-160 “Despite being highly speculative, our hypothesis may be supported...”)

p8: Line 188: I really think you need some form of wording supporting the statement of African origin here

Authors: The three cases were actually described in the supplementary text. We have now clarified this as follows "The three isolates concerned – which are detailed in the supplementary text section 'French autochthonous sporadic cases'– belonged to EUR1/AFR1 and probably originated from North Africa (n=2) or West Africa (n=1)."

p10: I am rather unconvinced that we really still need to be confirming that simple genotyping methods are not discriminatory in the light of genomic data

Authors: Ribotyping was extensively used in the past to type V. cholerae O1 isolates. However, there is a lack of studies comparing WGS and ribotyping data for V. cholerae O1 isolates. Here, we had an opportunity to show the limitations of the ribotyping method, particularly when dealing with isolates collected over a long period of time. We would therefore prefer to maintain this paragraph, but it could be moved to the supplementary material if the Editor prefers.

p11: I cant help but feel the AMR section would be strengthened by presenting a comprehensive genomic analysis of AMR genes in all the European strains and how this aligns with AST. This could be presented in a single annotated phylogeny

Authors: In accordance with the reviewer's suggestion, we have drawn a new figure (fig. 3) comparing antibiotic susceptibility data and resistome results, according to the phylogeny.

p31: On a personal level I would really like to see a classical rectangular visualisation of this timed phylogeny.

Authors: We have redrawn this phylogeny, which can now be visualized in a rectangular form.

I also wonder if some of the national outbreaks need amplified phylogenies as multi panel or supplementary figures to allow the reader to interrogate

Authors: An interactive tree containing all the genomes studied and linked to metadata that can be fully explored is now provided at <https://microreact.org/project/choleraeurope>. This link has been included into the manuscript.

Minor comments:

p8: Line 190-206: I personally dont like this style of short paragraphs and would like to see this merged to a single narrative paragraph

p10: Similarly the break in paragraphs at line 240 seems unnecessary

Authors: We have now reorganized this section into three main paragraphs. It would have been difficult for the reader to maintain attention over a single paragraph.